# Prediction Models for Elevated Cardiac Biomarkers from Previous Risk Factors and During the COVID-19 Pandemic in Residents of Trujillo City, Peru

**DOI:** 10.3390/diagnostics14222503

**Published:** 2024-11-08

**Authors:** Joao Caballero-Vidal, Jorge Luis Díaz-Ortega, Irma Luz Yupari-Azabache, Luz Angélica Castro-Caracholi, Juan M. Alva Sevilla

**Affiliations:** 1Escuela Profesional de Nutrición, Universidad César Vallejo, Trujillo 13001, Peru; jcaballerov@ucv.edu.pe (J.C.-V.); lcastroc@ucv.edu.pe (L.A.C.-C.); 2Institutos y Centros de Investigación, Universidad César Vallejo, Trujillo 13001, Peru; iyupari@ucv.edu.pe; 3Escuela Profesional de Medicina, Universidad César Vallejo, Trujillo 13001, Peru; jalvas@ucv.edu.pe

**Keywords:** natriuretic peptide, cardiac troponin I, metabolic syndrome, cholesterol, comorbidity, dyslipidemia

## Abstract

Cardiac troponin serum concentration is a marker of myocardial injury, but NT-pro BNP is a marker of myocardial insufficiency. The purpose of this study was to determine binary logistic regression models to verify the possible association of cardiovascular risk indicators, pre-pandemic history, the number of times participants were infected with SARS-CoV-2, and vaccination against these biomarkers. A total of 281 residents of Trujillo city (Peru) participated between September and December 2023. We found a high prevalence of abdominal obesity of 55.2%; glycemia > 100 m/dL in 53%; hypercholesterolemia in 49.8%; low HDL in 71.9%; and LDL > 100 mg/dL in 78.6%. A total of 97.5% were vaccinated against COVID-19, and 92.2% had three or more doses. Also, 2.5% had cTnI > 0.05 ng/mL, and 3.3% had NT-proBNP > 125 pg/mL. The number of COVID-19 infections versus cTnI > 0.05 ng/mL presented an OR = 3.513 (*p* = 0.003), while for NT-proBNP > 125 pg/mL, the number of comorbidities presented an OR = 2.185 (*p* = 0.025) and LDL an OR = 0.209 (*p* = 0.025). A regression model was obtained in which there is an association between a higher number of COVID-19 infections and elevated cTnI values and a model implying an association of the number of comorbidities and LDL with the NT-proBNP level in a direct and inverse manner, respectively. Both models contribute to the prevention of cardiac damage.

## 1. Introduction

Natriuretic peptides (NPs) are biochemical biomarkers of cardiac function, which correlate with the severity of heart failure (HF) and strongly predict the future development of HF [1]. The synthesis and secretion of NPs, predominantly atrial natriuretic peptides (ANPs) and brain-derived natriuretic peptides (BNPs), take place in response to myocardial stretch and fluid overload, while various stimuli have direct and indirect impacts on the production, accumulation, and secretion of NPs, such as ischemia/hypoxia, inflammation, hormones (catecholamines, aldosterone, renin), and growth factors (transforming growth factor-beta, vascular endothelial growth factor). Patients with asymptomatic or symptomatic HF may be classified as at risk of all-cause death and cardiovascular disease if they have elevated circulating levels of NT-proBNP > 125 pg/mL or >300 pg/mL, respectively [2].

Mild elevations in cardiac troponin levels are associated with the incidence of cardio-vascular disease and mortality. These increases, even within normal limits, are termed subclinical myocardial injury. Cardiac troponins also predict the risk of myocardial infarction, atrial fibrillation, and ischemic stroke. Cardiac troponins are associated with left ventricular remodeling and fibrosis and are considered sensitive indicators of cardiac pathophysiological processes that eventually manifest as clinical disease, such as heart failure [3].

Cardiovascular risk factors observed in patients include advanced age, male sex, and the presence of comorbidities such as hypertension, diabetes and obesity, cardiovascular disease, alterations in the lipid profile, and inadequate lifestyle habits such as smoking, and patients with these factors have been identified as groups vulnerable to COVID-19 morbimortality, which also extends a threat to patients who have survived COVID-19 [4].

Hyperglycemia, the primary event in diabetes mellitus (DM), does not alone confer the level of cardiovascular risk but is combined with a group of factors to impart a high risk for accelerating atherogenesis, including insulin resistance (IR), hypertension, obesity, and dyslipidemia, often occurring concomitantly and referred to as metabolic syndrome, which has been consistently shown to have a higher prevalence in diabetic patients than in healthy control populations [5].

Dyslipidemia is a major risk factor for the development of future cardiovascular events, including coronary heart disease, myocardial infarction, stroke, and peripheral vascular disease. However, studies have shown that when these conventional lipid parameters such as triglycerides, total cholesterol, or HDL and LDL lipoproteins remain apparently normal or moderately high, lipid ratios, such as the Castelli I (total cholesterol/HDL) and Castelli II (LDL/HDL) risk indices, and the triglyceride (TG)/HDL ratio are diagnostic alternatives for the prediction of cardiovascular events [6].

Elevated ventricular volumes and mass have also been found in recently recovered patients, and in others, an ongoing myocardial inflammatory process 3 months after COVID-19 diagnosis has been found [5]. Prolonged sequelae of COVID-19, due to the persistence of symptoms beyond 3 months, generate emotional problems such as anxiety, depression, and cardiopulmonary symptoms related to chest pain, difficulty breathing, and fatigue, and autonomic manifestations such as postural orthostatic tachycardia are common in patients who suffered from it [7].

In the context of COVID-19 mRNA vaccines, adverse cardiac events ranging from inflammation (such as pericarditis or myocarditis) to thrombosis and ischemia have been reported. Although some studies have reported on the frequency of cardiac complications following the administration of these vaccines, there has not been a large-scale observational study or a specific systematic review focused on mRNA vaccines yet. Because these adverse events are rare, most of the available articles are case reports and case series. A wide variety of adverse events and complications related to mRNA-based COVID-19 vaccines have been documented, which can affect both the nervous and cardiovascular systems. Examples of these events include acute myocardial infarction, pulmonary embolism, stroke, and venous thromboembolism [8].

Therefore, the present study attempts to evaluate a series of factors in which persons with a previous history of COVID-19 infection, risk indicators, and vaccination are possibly associated with cardiac risk with the use of two cardiac biomarkers, cardiac troponin I (cTnI) and NT-proBNP.

## 2. Materials and Methods

### 2.1. Type and Design of Research

The present study is a quantitative approach, non-experimental design, predictive-level study.

### 2.2. Population, Sample, and Sampling

The population consisted of employees of a private university, constituted by 2000 people, and former students of an emblematic educational institution of the city, together with their spouses, constituted by 3000 people, for a total of 5000. With this known population [9], we worked with a sample of 281 people, having a 5.68% error and 95% reliability, consisting of 169 (60%) people for the private university and 112 (40%) for the association of former students of the educational institution considered in the present study.

The sampling technique was non-probabilistic, accidental, and casual [10], in which the volunteers were informed by communication via e-mail, text messages to their cell phone, and/or Facebook by the director of their institution and registered by the researchers, using a previous orientative communication about the objective of the research and the biochemical analyses, reserving the date of their evaluation in a particular way.

People over 18 years of age that are healthy or have comorbidities such as hypertension, dyslipidemia, and diabetes were included, for an initial number of 300 people recruited. This study excluded those undergoing treatment with drugs such as antiplatelet agents or anticoagulants, generally given to patients diagnosed with cerebrovascular accident (CVA) and/or CVD established by thrombosis in whom cardiac marker values are elevated, especially NT-pro BNP; those taking antidepressants, antipsychotics, or corticoids were also excluded because these drugs promote metabolic syndrome since they influence obesity and the balance of lipid metabolism. Also not considered were those who presented pulmonary embolism, Alzheimer’s disease, pregnant women, people with some condition that prevented their nutritional evaluation, and those who practice long-distance running, the reason for the last group being excluded being related to a possible increase in troponin levels. Those with incomplete data on the data collection form were also excluded.

Finally, 281 participants remained for data analysis, the results of which are summarized in Figure 1. Additionally, having a positive COVID-19 diagnosis and having renal insufficiency were considered exclusion criteria; however, no cases of these were presented.

The data collection technique was non-experimental field observation, using a registration form as an instrument for general participant data such as age, sex, previous health history and vaccination against COVID-19, and finally data for physiological and biochemical characteristics in relation to cardiovascular risk factors: abdominal perimeter, systolic and diastolic blood pressure, glycemia, triglyceride concentration, HDL concentration, and atherogenicity indicators and cardiac biomarkers such as cTnI and NT-proBNP.

### 2.3. Evaluation of Lipid Profile and Atherogenic Indicators

The evaluation of the lipid profile was performed in the participants’ institutions bet-ween September and December 2023. Whole blood was obtained from the patients under aseptic conditions using the Vacutainer^®^ vacuum system in tubes with a clot activator of a 4 mL capacity. Once the sample was obtained, 35 μL of blood was collected immediately and placed in a vertical position in the central hole of the test strip previously inserted into the cholesterol monitoring equipment (Mission^®^, Acon Laboratories, San Diego, CA, USA).

The results of the lipid profile, total cholesterol, LDL, HDL, and triglycerides were recorded on the data collection form. Elevated values for total cholesterol (TC) were ≥200 mg/dL; triglycerides ≥ 150 mg/dL; LDL ≥ 100 mg/dL; and with respect to HDL, the values for cardiovascular risk in men and women were considered to be below 40 mg/dL and 50 mg/dL, respectively [11]. Based on the data obtained, atherogenicity indices were determined, whose risk values, according to Belalcazar [6], were as follows: TC/HDL values ≥ 5 indicate cardiovascular risk in males and ≥4.5 the established risk in females; LDL/HDL ≥ 3 risk in males and females; and the TG/HDL index in which values ≥ 3 denotes the risk of IR. Finally, according to Millan et al. [12], non-HDL cholesterol ≥ 130 mg/dL was considered to indicate cardiovascular risk.

### 2.4. Evaluation of Glycemia

In the case of glycemia, blood from the same finger used for the cholesterol measurement was used, placing it on the test strip previously inserted into a glucometer (Accu-Chek^®^ Performa Nano, Roche, Mannheim, Germany). Values higher than 100 mg/dL were considered to be abnormal fasting glucose levels according to the American Diabetes Association [13].

### 2.5. Evaluation of Blood Pressure and Abdominal Perimeter

For blood pressure, a digital blood pressure monitor (Riester^®^ Ri-Champion N, Jungingen, Germany) was used, and high blood pressure was considered when systolic blood pressure was ≥130 mmHg and/or diastolic blood pressure was ≥85 mmHg. In reference to nutritional status, abdominal perimeter was assessed with a metallic tape measure (Lufkin W606PM, Queretaro, Mexico), and abdominal obesity values were considered to be ≥94 cm in men and ≥88 cm in women [14].

### 2.6. Identification of Previous Antecedents of Health and Vaccination

Participants were previously asked if they had one or more comorbidities, which was determined by the consumption of medication for a given treatment in the last 12 months and whether they were still taking it; whether or not they had COVID-19; how long ago they had it; whether they had been in an intensive care unit; how many times they had COVID-19; whether they had been vaccinated; how many doses; and what type of vaccine (mRNA, DNA, or classical). This information was recorded on a data collection form.

### 2.7. Diagnosis with SARS-CoV-2 Antigen Testing

A rapid test kit for the detection of the SARS-CoV-2 antigen from Labnovation Technologies^®^, Guangdong, China was used, consisting of a dropper, nasal and oropharyngeal swab, buffer solution, and test cassette.

Prior to sampling, the analyst donned personal protective equipment, a swab was inserted through the patient’s nostrils into the nasopharyngeal area, and then the swab was rotated on its axis up to 5 times to obtain superficial cells of the respiratory epithelium that present the virus, and for oropharyngeal sampling, the participant was instructed to open their mouth to expose the tonsils and insert the swab to perform a swabbing motion from top to bottom 3 times. The samples on the swab were soaked in a buffer solution in the dropper using a circular motion, and 3 drops of the mixture were placed in the test cassette and allowed to react for 15 min.

For the interpretation of the results, the cassette has two bands, control (C) and test (T), with a negative test appearing as a colored line in the C band, while a positive test appears as a colored line in both the C and T bands, with the test invalidated if no color is present in the C band [15].

### 2.8. Evaluation of Cardiac Biomarkers

N-terminal pro-brain natriuretic peptide (NT-proBNP) and cTnI levels were assessed by time-resolved fluorescence immunoassay (TRFIA) using the LS-1100 dry immunofluorescence analyzer (Lansionbio^®^, Nanjing, China) [16,17].

Whole blood collected from patients for the lipid profile and glycemia was allowed to clot for 10 min and then centrifuged at 3000 rpm for 5 min to obtain serum. For NT-proBNP evaluation, 100 μL of serum was placed on the test strip with a micropipette, while for cTnI, the sample was previously diluted in buffer present in the test kit for diffusion to the antibody immobilized on the test strip. The antigen–antibody reaction time for the cTnI and NT-proBNP tests were 10 and 15 min at room temperature, respectively. The QR code of the test to be analyzed was placed in front of the scanner camera of the kit for setup. The test strip was inserted into the test slot of the immunofluorescence analyzer, and the result displayed on the instrument screen was printed.

In the case of NT-pro-BNP, an elevated value is defined as one greater than 125 pg/mL, according to the cut-off value used for adults under 75 years of age [16,18], and for cTnI, a value of 0.05 ng/mL was considered [17].

### 2.9. Statistical Analysis

The information from this study’s participants was collected on a data collection form and then entered into a Microsoft Excel spreadsheet. The SPSS version 27 statistical package was used for the corresponding processing, where descriptive statistical measures such as the mean, standard deviation, and inferential statistics were calculated to test the hypothesis. For bivariate analysis, the chi-square test was used to identify the relationship between qualitative variables, gamma test for the association between ordinal variables, Spearman correlation for the association between a quantitative variable and the ordinal qualitative variable, and the Mann–Whitney U test for quantitative variables. A multivariate analysis of binary logistic regression was also performed, establishing two models to identify factors that predict dependent variables [19].

### 2.10. Ethical Aspects

This study was approved by the Ethics Committee of the Professional School of Nutrition of the Universidad César Vallejo PI-CEI-NUTRICION-2023-007. The ethical principles of the Declaration of Helsinki were considered, and the participants were asked to give their voluntary consent to participate in this study. Each participant received in-formation on the objectives, basic protocols of the analyses to be performed, and knowledge of the researcher’s institutional affiliations in order to ensure their understanding of the information so that the participants could decide to accept or decline participation in this research.

## 3. Results

### 3.1. Biochemical Parameters, Anthropometric Parameters, and Previous History

Table 1 shows that there are characteristics such as abdominal perimeter, glycemia, blood pressure, cholesterol, HDL, the number of risk factors, TC/HDL, TG/HDL, and LDL/HDL that are associated with the sex of the persons analyzed (*p* < 0.05).

A detailed analysis of the before mentioned characteristics shows that the abdominal perimeter and glycemia of the female sex was mostly normal, and in the male sex, there is a greater tendency of abdominal obesity and elevated fasting glycemia; blood pressure in both sexes is <130/85 mmHg, cholesterol is higher in women, HDL in both sexes is low, most have less than three risk factors, and 45.9% have metabolic syndrome according to the ATPIII criteria, being more prevalent in men than in women. Likewise, we found the same proportion of women with elevated (>3) and normal (<3) values in terms of TC/HDL, but in men, the majority have elevated and cardiovascular risk values, and there is a higher proportion of TG/HDL and LDL/HDL in cardiovascular risk values (>3) in the men’s group.

Regarding quantitative variables, there were no differences in the number of comorbidities and the number of COVID-19 infections between men and women.

The prevalence of an elevated abdominal perimeter and elevated glycemia was observed in more than 50% of the participants, as well as the predominance of hypercholesterolemia versus hypertriglyceridemia, and in the case of lipoproteins, there was a high prevalence of LDL with values above 100 mg/dL in about 79% and low HDL in a proportion of 72%. Regarding the atherogenic indicators, a higher proportion of non-HDL cholesterol was found, followed by LDL/HDL and total cholesterol/HDL.

### 3.2. Cardiac Troponin I

Table 2 shows the anthropometric and biochemical characteristics according to cTnI levels, where only blood pressure and the number of COVID-19 infections were significantly related to cTnI levels (*p* < 0.05). A total of 57.6% were infected with COVID-19, of which 2.1% reached the intensive care unit. Likewise, a proportion of 2.5% of participants with cTnI levels ≥ 0.05 ng/mL was observed; of these, 2.1% were infected with COVID-19 at least once.

### 3.3. NT-proBNP

Table 3 shows that of all the anthropometric and biochemical characteristics analyzed, only LDL and the number of comorbidities were significantly related to cardiac NT-proBNP values (*p* < 0.05). Likewise, the percentage of persons with an NT-proBNP concentration ≥ 125 pg/mL corresponds to 3.2%, of which 1.4% presented LDL values ≥ 100 mg/dL, and 1.5% presented COVID-19 infection at least once.

### 3.4. Regression Model for cTnI

Table 4 shows the binary logistic regression model for cTnI in residents of the city of Trujillo, entering only the variable of the number of COVID-19 infections, which indicates that there is a 3.5 times greater probability that a person with a greater number of COVID-19 infections will have a diagnosis of elevated cTnI (>0.05 ng/mL).

The model was established as follows:$$p(y)=\frac{1}{{1+e}^{\left(5.150-1.256x1\right)}}$$
where *y*: a diagnosis of elevated cardiac cTnI; *X*_1_: the number of COVID-19 infections. The prognostic percentage is high, and the model is significant (*p* < 0.05); however, it only correctly predicts the diagnosis of people with a cTnI level less than 0.05 ng/mL; this is because no people with a cTnI level greater than 0.05 ng/mL were found in the sample analyzed. Thus, if *X*_1_ = 3 COVID-19 infections, the maximum value found among the participants, there is a 20% probability that the cTnI level is above 0.05 ng/mL, whereas in the case of four infections, the probability approaches 50% for such a situation.

### 3.5. Regression Model for NT-proBNP

Table 5 shows the binary logistic regression model for cardiac NT-proBNP in residents of the city of Trujillo, including the variables LDL and the number of comorbidities. The LDL variable contains a coefficient with a negative sign, indicating that it apparently acts as a protective factor, a contradictory aspect that will be explained later, and the number of comorbidities as a risk factor, indicating that when there are more comorbidities in people, the NT-proBNP level tends to be higher.

The model was established as follows:$$p(y)=\frac{1}{{1+e}^{\left(2.769+1.565x1-0.782x2\right)}}$$
where *y*: a diagnosis of elevated NT-proBNP; *X*_1_: LDL level; *X*_2_: the number of comorbidities. The prognostic percentage is high, and the model is significant (*p* < 0.05), but it only correctly predicts the diagnosis of people with an NT-proBNP level less than 125 pg/mL, as no people with an NT-proBNP level greater than 125 pg/mL were found in the analyzed sample. For *X*_1_ = 0 with LDL < 100 mg/dL and *X*_2_ = 3 comorbidities, a 40% probability of NT-proBNP values above 125 pg/mL can be deduced. If *X*_1_ = 1, for an LDL level ≥ 100mg/dL and *X*_2_ = 0 for the number of comorbidities is 0, the probability of an NT-proBNP level above 125 pg/mL is very low, corresponding to 0.01%; however, 99.99% is predicted for an NT-proBNP level < 125 pg/mL, which would occur, for example, for the average LDL in the participant group corresponding to 131 mg/dL.

## 4. Discussion

In relation to Table 1, the proportions of abdominal obesity and dyslipidemia found in the inhabitants of the city of Trujillo are also high in other cities of Peru, such as in the study by Barboza [20] in the inhabitants of the city of Ayacucho, in which men have a higher prevalence, approximately 57.4% with dyslipidemia and 57.7% with obesity, as well as 68.8% with DM2 and 63% with arterial hypertension (HT), compared to women, in whom obesity and dyslipidemia stand out, followed by HT and DM2 with prevalences of 42.3%, 42.6%, 36%, and 31%, respectively. DM2 was not measured in this study, but elevated fasting glucose was found in more than 50% of participants.

High levels of dyslipidemia were also found in the study by Ruiz and Farro [21] in patients admitted to a medical center in the city of Chiclayo; however, in the present study, there was a higher prevalence of both high LDL and low HDL levels.

The variables that were related to the cTnI level shown in Table 2 are in agreement with the study by Welsh et al. [22] who found strong positive associations for cTnI with systolic blood pressure; however, in their study, they also found associations with age, male sex, body mass index, and the use of cholesterol and blood pressure medications in a larger sample of Scottish participants identified over 4 years.

A longitudinal study in Norway showed that participants with relative decreases in cTnI were more often younger and female and had lower blood pressure and BMI. Participants with relative increases in cTnI were more often older men and had higher systolic blood pressure [23]. In the present study, we could not establish comparisons in the highest cTnI levels by age and sex, because of the few cases presented for TnI ≥ 0.05 ng/mL.

In our study, we identified a new factor associated with elevated cTnI levels in relation to the number of COVID-19 infections, in the context of the pandemic. It was observed that cTnI levels above 0.04 ng/mL and up to 0.39 ng/mL were present in 67.7% of patients with severe COVID-19, compared to 18.5% of patients who presented values below 0.04 ng/mL in a hospital setting [24]. This aspect is attenuated in a community and post-pandemic setting, as observed in the inhabitants of the city of Trujillo (Peru); although there are a few people who could have developed cardiac lesions due to COVID-19, it is still important to rule them out preventively by other confirmatory tests such as imaging studies.

SARS-CoV-2 targets angiotensin-converting enzyme 2 (ACE2) receptors. Although the lung is the primary target of coronavirus infection, the widespread presence of ACE2 receptors in several organs may have an impact on the cardiovascular system. This suggests a direct mechanism of action in cardiac tissue, which requires constant monitoring [25].

Myocarditis, which can arise as a result of COVID-19, is based on interleukin-6 (IL-6) stimulation and the resulting activation of a cytokine storm, in addition to direct damage to the heart [26].

The correlation between NT-proBNP and LDL levels observed in Table 3 was also found in the study by Spannella et al. [27] who also determined a negative correlation between NT-proBNP and total cholesterol and also non-HDL cholesterol. As in the present study, they also found no correlation between NT-proBNP concentration and both HDL and triglyceride concentration.

The present study was unable to determine the association between plasma NT-proBNP and other risk factors such as sex, age, glycemia, blood pressure, lipid profile components, atherogenic indices, COVID-19 vaccination, metabolic syndrome risk factors, and the number of infections. This differs from other studies such as that of Tanaka et al. [28] in which they managed to observe a positive association with age, female sex, systolic pressure, hypertension and HDL-c, and smoking, but smoking and other lifestyle habits were not considered in the binary logistic regression analysis.

In initial studies of patients with COVID-19, it has been observed that about 20–30% have elevated cardiac troponin levels, indicating damage to heart tissue. During SARS-CoV-2 infection, this damage may be due to factors such as immune response, hypercoagulability with thrombotic events, or myocardial ischemia. The mechanisms behind SARS-CoV-2 myocarditis are not yet fully understood, but a strong hypothesis suggests that cytotoxicity mediated by an excessive immune response may be the cause. Most cases of myocarditis in patients with COVID-19 do not come with the detection of viral RNA in cardiac tissue [29]. In patients with evidence of myocardial damage by troponin elevation, only approximately 50% of cardiac magnetic resonance (CMR) images show pathological findings. Of these CMR images, about 27% are compatible with myocarditis, while 22% are associated with ischemic heart disease [29]. 

Following vaccination against COVID-19 with mRNA vaccines, there are several hypotheses about the mechanisms involved in cardiac conditions, but there are currently insufficient data to confirm or refute any of them. However, vaccination and the number of doses were not associated with cTnI in the present study.

The incidence of myocarditis in the case of reinfections by SARS-CoV-2 is unknown, and its characteristics during the initial infection are also unknown [29]. In this study, we consider it important that in people with mild COVID-19, the number of times of infection by the virus should be monitored, and any established cardiac condition should be ruled out.

A multiple regression model obtained by Zhu et al. [30] found a greater number of associations of NT-proBNP with other variables; thus, they found a positive association with systolic blood pressure, fasting plasma glucose, and total cholesterol. NT-proBNP levels were inversely associated with diastolic blood pressure, abdominal perimeter, triglycerides, and Low-density lipoprotein (LDL) in all participants in their study. LDL, whose negative ratio in the binary logistic regression model apparently indicated that it was a protective factor, should be analyzed with great care. In recent years, new metabolic functions of NPs have been identified, including the activation of lipolysis, lipid oxidation, and mitochondrial respiration. NPs may also ameliorate lipid-induced IR through improvements in lipid oxidation at the hepatic and muscular levels [31].

This would mean that an increase in NT-proBNP levels would be associated with low concentrations in the atherogenic components of the lipid profile, including LDL, as observed in the logistic regression model in Table 5. The reasons why NT-proBNP values are decreased in the metabolic syndrome are unknown, and in this pathology, most patients have an altered lipid profile with elevated values of total cholesterol, LDL, and triglycerides. This makes sense since in the participating population of the city of Trujillo, 78,6% have elevated LDL levels, while NT-proBNP levels below 50 pg/mL represent 76.9%, and between 50 and 125 pg/mL corresponds to 19.9%; the data are not shown in the results, but we have them, and they allow us to corroborate this inverse relationship between these parameters.

Likewise, the research of Sanchez et al. [32] demonstrated the inverse relationship between the LDL and NT-proBNP levels in a biphasic behavior, where the decrease in LDL concentration, from 130 mg/dL to 115 mg/dL, when NT-proBNP concentration is increased up to approximately 50 pg/mL, is part of the natural protective response of the peptide. If NT-proBNP values are between 50 and 125 pg/mL, there is a stabilization in the variation in LDL concentration, with a plateau of this lipoprotein being observed around 114 mg/dL, while above 125 pg/mL and up to 300 pg/mL, it is slightly reduced to 110 mg/dL. This is why the logistic regression model for NT-proBNP can only predict up to the threshold of 125 pg/mL, and we try to explain this with the physiological levels of LDL to be reached corresponding to <100 mg/dL.

It is important to emphasize that substantially elevated levels of NT-proBNP, potentially due to the presence of subclinical cardiovascular disease, can no longer influence blood lipids and therefore neither BMI nor IR, which means that there is a state of low response to the natriuretic peptide and therefore loss of its inverse association under physiological conditions, in which the peptide has a greater preponderance to the remodeling of the heart [32].

It is important to have a diet rich in fruits, vegetables, and whole grains and decrease the intake of foods rich in saturated fats, such as whole meats and dairy, and sweets and sugary drinks, and regular physical activity is important to reduce LDL cholesterol and cardiovascular risk [33]. Diets rich in fruits and vegetables administered have been associated with lower levels of biomarkers of subclinical heart damage and stress in adults without pre-existing cardiovascular disease [34]. These healthy lifestyles would contribute mainly to controlling LDL concentrations and helping the physiological functions of NT-proBNP to avoid reaching situations of severe cardiac damage.

One limitation of this study is that the results and statistical models estimated, due to the type of sampling, have a certain degree of difficulty to be generalized to the entire population of Trujillo; however, because they worked with two institutions that are attended by people from different districts of the city, they can support in some way what was found in reality.

Regarding the measurement of cTnI, one limitation was the lack of a high-sensitivity assay for this marker, since immunofluorescence equipment does not yet have such a test. This could detect a larger number of probable cases of cardiac involvement, since the high-sensitivity troponin assay can detect levels approximately 10 times lower than those of the conventional assay [35]. This aspect can also be taken into account in future studies involving the random measurement of cTn in patients without clinical suspicion of acute cardiovascular events, which would be useful because of its correlation with the presence of subclinical cardiovascular dysfunction [36] and, therefore, to establish predictive models for such a situation.

Another limitation would be the possible cross-reactivity between NT-proBNP and proBNP in the NT-proBNP assay, and this could overestimate their values [32], which may influence the linear association with the different lipid variables to be correlated.

Conversely, as a strength of this study, we can mention that appropriate statistical methods were used that allowed us to analyze bivariate and multivariate variables. For the multivariate variables, we only analyzed the most relevant variables that could predict the behavior of the altered markers and that were explained by the contributions of other previously mentioned studies. Another advantage of the present work is its application in persons without any clinical symptoms or who did not present a diagnosis of disease with cardiovascular involvement, which reduces the bias of the results, especially in the regression models obtained for the elevated cardiac markers. Therefore, health personnel should take into account the number of COVID-19 infections for cTnI and LDL concentration, as well as the number of comorbidities for NT-proBNP, when recommending preventive actions and keeping the population under surveillance in order to avoid heart disease, especially in special situations of a pandemic caused by a respiratory virus, and thus counteract mortality.

## 5. Conclusions

A regression model was determined for elevated cTnI values in which only the number of COVID-19 infections was directly proportional; however, longitudinal studies are needed to improve this model in order to prevent myocardial lesions and myocardial failure and to rule them out with any of the imaging techniques.

In the case of the prediction model of elevated NT-proBNP values, LDL concentration was inversely associated with a greater number of comorbidities and, directly, with a greater number of comorbidities. This prediction model determines the importance of the balancing role of NT-proBNP against comorbidities related to cardiovascular disease, such as dyslipidemia, insulin resistance, and type 2 diabetes mellitus, in which LDL concentration may be increased.

## Figures and Tables

**Figure 1 diagnostics-14-02503-f001:**
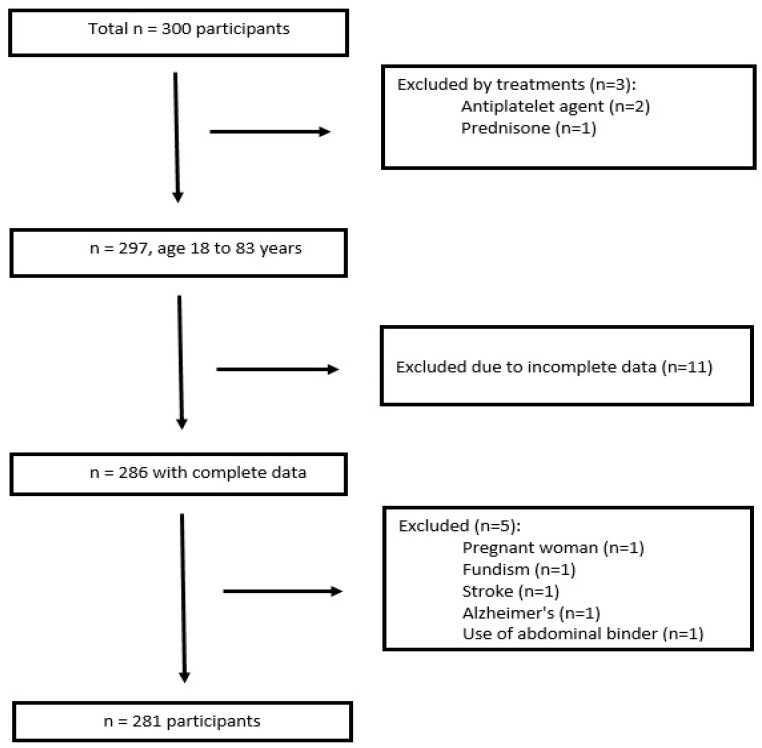
Flow diagram of subject inclusion and exclusion.

**Table 1 diagnostics-14-02503-t001:** Physiological and biochemical characteristics and previous history according to sex in residents of city of Trujillo, Peru.

Characteristics		Sex				Total	%	Sig.
		Female	%	Male	%			
Age (years)	≥50	74	26.3	44	15.7	118	42.0	0.52
	<50	96	34.2	67	23.8	163	58.0	
Abdominal perimeter (cm)	F ≥ 88; M ≥ 94	83	29.5	72	25.6	155	55.2	0.008
	F < 88; M < 94	87	31.0	39	13.9	126	44.8	
Glycemia (mg/dL)	≥100	78	27.8	71	25.3	149	53.0	0.003
	<100	92	32.7	40	14.2	132	47.0	
Blood pressure (mmHg)	≥140/90	14	5.0	12	4.3	26	9.3	0.018
	130/85–139/89	12	4.3	19	6.8	31	11.0	
	<130/85	144	51.2	80	28.5	224	79.7	
Cholesterol (mg/dL)	≥200	99	35.2	41	14.6	140	49.8	0.000
	<200	71	25.3	70	24.9	141	50.2	
Triglycerides (mg/dL)	≥150	54	19.2	45	16.0	99	35.2	0.132
	<150	116	41.3	66	23.5	182	64.8	
HDL (mg/dL)	F < 50; M < 40	113	40.2	89	31.7	202	71.9	0.012
	F ≥ 50; M ≥ 40	57	20.3	22	7.8	79	28.1	
LDL (mg/dL)	≥100	138	49.1	83	29.5	221	78.6	0.2
	<100	32	11.4	28	10.0	60	21.4	
Number of risk factors	≥3	28	10.0	34	12.1	62	22.1	0.00
	3	30	10.7	37	13.2	67	23.8	
	<3	112	39.9	40	14.2	152	54.1	
TC/HDL	F ≥ 4.5; M ≥ 5	85	30.2	76	27.0	161	57.3	0.00
	F < 4.5; M < 5	85	30.2	35	12.5	120	42.7	
TG/HDL	≥3	71	25.3	75	26.7	146	52.0	0.00
	<3	99	35.2	36	12.8	135	48.0	
LDL/HDL	≥3	87	31.0	87	31.0	174	61.9	0.00
	<3	83	29.5	24	8.5	107	38.1	
COL No HDL (mg/dL)	≥130	127	45.2	82	29.2	209	74.4	0.88
	<130	43	15.3	29	10.3	72	25.6	
Intensive care unit	Yes	4	1.4	2	0.7	6	2.1	0.76
	No	166	59.1	109	38.8	275	97.9	
Vaccines	No	5	1.8	2	0.7	7	2.5	0.55
	Yes	165	58.7	109	38.8	274	97.5	
Dosage	≥3	154	54.8	105	37.4	259	92.2	0.22
	<3	16	5.7	6	2.1	22	7.8	
Total		170	60.5	111	39.5	281	100.0	
Number of comorbidities		0.26 ± 0.61		0.24 ± 0.61		0.25 ± 0.60		0.83 ^a^
Number of COVID-19 infections		0.74 ± 0.78		0.77 ± 0.77		0.75 ± 0.77		0.6 ^a^

Note: F (female); M (male). Chi-square test of association was used, and in ^a^, Mann–Whitney U test was used.

**Table 2 diagnostics-14-02503-t002:** Physiological and biochemical characteristics and previous history according to cTnI in residents of city of Trujillo, Peru.

Characteristics		cTnI (ng/mL)				Total	%	Sig.
		<0.05	%	≥0.05	%			
Age (years)	≥50	116	41.3	2	0.7	118	42.0	0.73
	<50	158	56.2	5	1.8	163	58.0	
Genre	Female	165	58.7	5	1.8	170	60.5	0.94
	Male	109	38.8	2	0.7	111	39.5	
Abdominal perimeter (cm)	F ≥ 88; M ≥ 94	151	53.7	4	1.4	155	55.2	1.00
	F < 88; M < 94	123	43.8	3	1.1	126	44.8	
Glycemia (mg/dL)	≥100	146	52.0	3	1.1	149	53.0	0.87
	<100	128	45.6	4	1.4	132	47.0	
Blood pressure (mmHg)	≥140/90	26	9.3	0	0.0	26	9.3	0.01 ^a^
	130/85–139/89	31	11.0	0	0.0	31	11.0	
	<130/85 mmHg	217	77.2	7	2.5	224	79.7	
Cholesterol (mg/dL)	≥200	137	48.8	3	1.1	140	49.8	1.00
	<200	137	48.8	4	1.4	141	50.2	
Triglycerides (mg/dL)	≥150	97	34.5	2	0.7	99	35.2	1.00
	<150	177	63.0	5	1.8	182	64.8	
HDL (mg/dL)	F < 50; M < 40	195	69.4	7	2.5	202	71.9	0.21
	F ≥ 50; M ≥ 40	79	28.1	0	0.0	79	28.1	
LDL (mg/dL)	≥100	216	76.9	5	1.8	221	78.6	0.10
	<100	58	20.6	2	0.7	60	21.4	
Number of risk factors	>3	60	21.4	2	0.7	62	22.1	0.89 ^a^
	3	66	23.5	1	0.4	67	23.8	
	<3	148	52.7	4	1.4	152	54.1	
TC/HDL	F ≥ 4.5; M ≥ 5	157	55.9	4	1.4	161	57.3	1.00
	F < 4.5; M < 5	117	41.6	3	1.1	120	42.7	
TG/HDL	≥3	142	50.5	4	1.4	146	52.0	1.00
	<3	132	47.0	3	1.1	135	48.0	
LDL/HDL	≥3	170	60.5	4	1.4	174	61.9	1.00
	<3	104	37.0	3	1.1	107	38.1	
COL No HDL (mg/dL)	≥130	205	73.0	4	1.4	209	74.4	0.54
	<130	69	24.6	3	1.1	72	25.6	
Intensive care unit	Yes	6	2.1	0	0.0	6	2.1%	1.00
	No	268	95.4	7	2.5	275	97.9	
Vaccines	No	7	2.5	0	0.0	7	2.5	1.00
	Yes	267	95.0	7	2.5	274	97.5	
Dose	≥3	252	89.7	7	2.5	259	92.2	0.95
	<3	22	7.8	0	0.0	22	7.8	
Number of comorbidities	0	223	79.4	6	2.1	229	81.5	0.74 ^b^
	1	38	13.5	1	0.4	39	13.9	
	2	7	2.5	0	0.0	7	2.5	
	3	6	2.1	0	0.0	6	2.1	
Number of COVID-19 infections	0	118	42.0	1	0.4	119	42.3	0.01 ^b^
	1	120	42.7	2	0.7	122	43.4	
	2	29	10.3	2	0.7	31	11.0	
	3	7	2.5	2	0.7	9	3.2	

Note: F (female); M (male); ^a^ gamma test was used, ^b^ Spearman correlation, and chi-square in others.

**Table 3 diagnostics-14-02503-t003:** Physiological and biochemical characteristics and previous history according to NT-proBNP in inhabitants of city of Trujillo, Peru.

Characteristics		NT-proBNP (pg/mL)				Total	%	Sig.
		<125	%	≥125	%			
Age (years)	≥50	112	39.9	6	2.1	118	42.0	0.23
	<50	160	56.9	3	1.1	163	58.0	
Genre	Female	165	58.7	5	1.8	170	60.5	0.76
	Male	107	38.1	4	1.4	111	39.5	
Abdominal perimeter (cm)	F ≥ 88; M ≥ 94	151	53.7	4	1.4	155	55.2	0.75
	F < 88; M < 94	121	43.1	5	1.8	126	44.8	
Glycemia (mg/dL)	≥100	145	51.6	4	1.4	149	53.	0.85
	<100	127	45.2	5	1.8	132	47.0	
Blood pressure (mmHg)	≥140/90	24	8.5	2	0.7	26	9.3	0.08 ^a^
	130/85–139/89	28	10.0	3	1.1	31	11.0	
	<130/85	220	78.3	4	1.4	224	79.7	
Cholesterol (mg/dL)	≥200	138	49.1	2	0.7	140	49.8	0.18
	<200	134	47.7	7	2.5	141	50.2	
Triglycerides (mg/dL)	≥150	98	34.9	1	0.4	99	35.2	0.24
	<150	174	61.9	8	2.8	182	64.8	
HDL (mg/dL)	F < 50; M < 40	194	69.0	8	2.8	202	71.9	0.44
	F ≥ 50; M ≥ 40	78	27.8	1	0.4	79	28.1	
LDL (mg/dL)	≥100	217	77.2	4	1.4	221	78.6	0.03
	<100	55	19.6	5	1.8	60	21.4	
Number of risk factors	>3	59	21.0	3	1.1	62	22.1	0.49 ^a^
	3	65	23.1	2	0.7	67	23.8	
	<3	148	52.7	4	1.4	152	54.1	
TC/HDL	F ≥ 4.5; M ≥ 5	158	56.2	3	1.1	161	57.3	0.26
	F < 4.5; M < 5	114	40.6	6	2.1	120	42.7	
TG/HDL	≥3	142	50.5	4	1.4	146	52.0	0.91
	<3	130	46.3	5	1.8	135	48.0	
LDL/HDL	≥3	170	60.5	4	1.4	174	61.9	0.45
	<3	102	36.3	5	1.8	107	38.1	
COL No HDL (mg/dL)	≥130	205	73.0	4	1.4	209	74.4	0.09
	<130	67	23.8	5	1.8	72	25.6	
Intensive care unit	Yes	6	2.1	0	0.0	6	2.1	1.00
	No	266	94.7	9	3.2	275	97.9	
Vaccines	No	7	2.5	0	0.0	7	2.5	1.00
	Yes	265	94.3	9	3.2	274	97.5	
Dose	≥3	250	89.0	9	3.2	259	92.2	0.80
	<3	22	7.8	0	0.0	22	7.8	
Number of comorbidities	0	224	79.7	5	1.8	229	81.5	0.03 ^b^
	1	37	13.2	2	0.7	39	13.9	
	2	6	2.1	1	0.4	7	2.5	
	3	5	1.8	1	0.4	6	2.1	
Number of COVID-19 infections	0	114	40.6	5	1.8	119	42.3	0.44 ^b^
	1	119	42.3	3	1.1	122	43.4	
	2	30	10.7	1	0.4	31	11.0	
	3	9	3.2	0	0.0	9	3.2	

Note: F (female); M (male); ^a^ gamma test was used, ^b^ Spearman correlation, and chi-square in all others.

**Table 4 diagnostics-14-02503-t004:** Binary logistic regression model for cTnI in residents of city of Trujillo, Peru.

	B	Standard Error	Wald	gL	Sig.	Exp(B)	95% C.I. for EXP(B)	
							Inferior	Superior
Number of COVID-19 infections	1.256	0.418	9.053	1	0.003	3.513	1.550	7.963
Constant	−5.150	0.791	42.413	1	0.000	0.006		
Cox and Snell’s R-squared: 0.03; Nagelkerke R-squared: 0.15								
Overall predicted percentage (only for cTnI < 0.05 ng/mL): 97.5%.								

Note: SPSS version 27, Wald forward method.

**Table 5 diagnostics-14-02503-t005:** Binary logistic regression model for cardiac NT-proBNP in residents of city of Trujillo, Peru.

	B	Standard Error	Wald	gL	Sig.	Exp(B)	95% C.I. for EXP(B)	
							Inferior	Superior
LDL	−1.565	0.700	4.995	1	0.025	0.209	0.053	0.825
Number of Comorbidities	0.782	0.349	5.018	1	0.025	2.185	1.103	4.331
Constante	−2.769	0.533	26.997	1	0.000	0.063		
Cox and Snell’s R-squared: 0.03; Nagelkerke R-squared: 0.13								
Overall predicted percentage (only for NT-ProBNP < 125 pg/m): 96.8%								

Note: SPSS version 27, Wald forward method.

## Data Availability

The data are not publicly available due to privacy issues (patient data). Data can be obtained from the corresponding author on justified request.

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
