# Peer review of "Prediction Models for Elevated Cardiac Biomarkers from Previous Risk Factors and During the COVID-19 Pandemic in Residents of Trujillo City, Peru"

_diagnostics, 2024, doi:10.3390/diagnostics14222503_

Round 1

Reviewer 1 Report

Comments and Suggestions for Authors

First of all, I congratulate the authors for the topic chosen in their manuscript regarding the identification of predictive models of cardiovascular risk factors in patients who have gone through SARS CoV-2 infection.

Unfortunately, the COVID-19 disease has presented a great challenge to medical professionals with many currently unknown data. This fact underlines the importance of high-quality scientific research in the fight against this disease and in the conditions where medical books are written in real time at "the patient's head".

I also note the originality of this article. Clear, scientifically relevant manuscript, with a design that methodically follows rigorous statistical analysis, scientifically argued results and conclusions.

In another order, I recommend the authors to review the data in the tables, where small errors have crept in (such as: abdominal perimeter F≥88; M≥94 and F<88; M≥94 - I think you wanted to write F< 88; M<94).

I recommend the authors to rewrite the conclusions, especially regarding troponin (for NTproBNP I have no comments).

Author Response

"Please see the attachment." in the box if you only upload an attachment.

Reviewer 2 Report

Comments and Suggestions for Authors

I have received for review an original research article entitled “Prediction Models for High Cardiac Biomarkers in Residents of 2 Trujillo City, Peru” prepared by Joao Caballero-Vidal et al., which had been submitted to Diagnostics (IF=3.0). Cardiovascular diseases are one of the most important public health problems worldwide constituting the leading cause of morbidity and mortality worldwide. Developing of knowledge about cardiovascular diseases as well as diagnostic tools used in the care on patients with cardiovascular diseases are therefore of crucial importance. In my opinion, presented manuscript touches an important issue and it represents some scientific value but significant modification are required which can further improve its value and attractiveness. I present my suggestions below.

1)     In the abstract it has been written: “Cardiac troponin I (cTnI) and NT-pro BNP are considered biomarkers for detecting suspected cardiac injury”. It must be corrected. Cardiac troponin serum concentration is a marker of myocardial injury but NT-pro BNP is a marker of myocardial insufficiency.

2)     In my opinion the title should be changed because it reflects the purpose of this paper not well. It suggests relation between cardiac biomarkers and typically studied cardiovascular risk factors, whereas an important part of this research is association between cardiac biomarkers and such aspects as COVID-19 history and vaccination what was very surprising for me.

3)     Although the introduction is generally quite well prepared, some aspects should be developed. In my opinion, it would be worth to mention additionally that cardiac troponin serum concentration randomly measure (without a clinical suspicion of acute cardiovascular event) was shown to be correlated with features of subclinical cardiovascular dysfunction (10.3390/jpm14030230).

4)     Why people taking namely antiplatelet agents and prednisone were excluded? Any other drugs has been also considered as an exclusion criterion? All exclusion criteria should be given point by point.

5)     Strengths and limitations of the study should be discussed more thoroughly.

6)     The section of conclusions should be significantly broaden. Conclusions cannot be consisted of only one sentence.

7)     References should be prepared fully in accordance with requirements of MDPI Publishing House.

8)     Small grammatical, stylistic, and editorial mistakes should be corrected. 

Comments on the Quality of English Language

Small grammatical, stylistic, and editorial mistakes should be corrected. 

Author Response

(The authors gave the same response as above.)

Round 2

Reviewer 2 Report

Comments and Suggestions for Authors

I have received for review a revised version of the original research article entitled “Prediction Models for High Cardiac Biomarkers in Residents of 2 Trujillo City, Peru” prepared by Joao Caballero-Vidal et al., which had been submitted to Diagnostics (IF=3.0). The paper has been significantly improved. I have no further critical comments. I recommend it for publication in its current form. Thank you for the invitation to prepare a review report of this article. I wish good luck for the Authors and further success in their scientific work.